# Who Gets the Reward & Who Gets the Blame? Evaluation-Aligned Training Signals for Multi-LLM Agents

## Abstract

Large Language Models (LLMs) in multi-agent systems (MAS) have shown promise for complex tasks, yet current training methods lack principled ways to connect system-level evaluation with agent- and message-level learning. We propose a theoretical framework that unifies cooperative game–theoretic attribution with process reward modeling to transform *system evaluation* $\rightarrow$ *agent credit* $\rightarrow$ *response-level signals*. Unlike prior approaches that rely only on attribution (Shapley) or step-level labels (PRM), our method produces *local, signed, and credit-conserving signals*. In success cases, Shapley-based credit assignment fairly allocates outcomes across agents and is refined into per-message rewards that promote cooperation while discouraging redundancy or sabotage; in failure cases, first-error localization yields *repair-aware preferences* that penalize harmful steps while rewarding corrective attempts. The resulting signals are bounded, cooperative, and directly compatible with reinforcement- or preference-based post-training, providing a unified and auditable pathway from global evaluation to local supervision in LLM multi-agent training. Our contribution is conceptual: we present a theoretical foundation and training signals, leaving empirical validation for future work.

## 1 Introduction

Multi-Agent Systems (MAS) built from Large Language Models (LLMs) are emerging as a powerful paradigm for complex tasks. *Agents* in this context may be LLMs, ML models, or tools (Ye et al., 2025; Wang et al., 2025b; He et al., 2024; Yang et al., 2024; Team, 2025), each taking structured inputs and producing outputs such as text, code, or tool calls. Unlike classical RL agents with compact, repetitive action spaces (Bamford & Ovalle, 2021; Li et al., 2021a; Hubert et al., 2021; Yue et al., 2020), LLM agents act in high-dimensional spaces with diverse, often unique responses (Ye et al., 2025; Lu et al., 2025; Li et al., 2025a; 2024b; He et al., 2024). This flexibility enables rich collaboration but also introduces fragility: errors can cascade to derail workflows (Cemri et al., 2025; He et al., 2025b; Huang et al., 2025; Owotogbe, 2025; Lin et al., 2025a), repair loops inflate runtime and costs (Cemri et al., 2025; He et al., 2025b; Owotogbe, 2025; Bo et al., 2024; Zhang et al., 2024b), and repeated coordination failures reduce reliability (Cemri et al., 2025; Huang et al., 2025; Owotogbe, 2025; Motwani et al., 2024; Nagpal et al., 2025). These challenges make *evaluation-aligned training* essential: system-level evaluation (e.g., success/failure, rubric scores, or process-based feedback) must guide both agent- and response-level learning in a fair, efficient, and auditable way.

Such alignment is well established for single LLMs. Post-training methods including RLHF (Ouyang et al., 2022), DPO (Rafailov et al., 2023), KTO (Ethayarajh et al., 2024), and GRPO (Luo et al., 2024) refine pretrained models with outcome- or process-level feedback, improving instruction following and alignment with human preferences across reasoning, summarization, and benchmark tasks (Bai et al., 2022; Askell et al., 2021; Lightman et al., 2023; Setlur et al., 2024; Zhang et al., 2024a). However, these approaches are inherently limited to the *single-agent* setting: evaluation signals map directly to one trajectory, with gradients or preferences flowing through a single model. In LLM-based MAS, by contrast, evaluation applies only at the system level, and attribution must cross multiple agents and steps (Li et al., 2024b; He et al., 2025a; Cemri et al., 2025), leaving single-agent

post-training methods ill-suited to this setting (Askell et al., 2021; Bai et al., 2022; Rafailov et al., 2023; Ethayarajh et al., 2024; Chan et al., 2024).

A natural source of inspiration is multi-agent reinforcement learning (MARL), where credit assignment has long been studied (Lowe et al., 2017; Foerster et al., 2018; Sunehag et al., 2018; Rashid et al., 2018; Böhmer et al., 2020; Arjona-Medina et al., 2019). These works show that dividing system rewards among agents or timesteps can drive cooperation, yet they rely on assumptions—low-dimensional, repetitive actions and dense numeric rewards—that do not hold in high-dimensional language settings (Li et al., 2024b; Yang et al., 2024; He et al., 2025a; Cemri et al., 2025; Park et al., 2023). Concretely, value factorization and coordination graphs assume discrete, repeatable state–action pairs (Ma et al., 2023; Park et al., 2023; Sharma et al., 2023; Yang et al., 2024; Li et al., 2024b); single-agent post-training methods lack mechanisms for distributing credit across agents and responses (Askell et al., 2021; Bai et al., 2022; Rafailov et al., 2023; Ethayarajh et al., 2024; Chan et al., 2024); and many RL methods presuppose online interaction and dense rewards, while LLM MAS often rely on offline logs and coarse evaluations (Ouyang et al., 2022; Lightman et al., 2023; Setlur et al., 2024; Zhang et al., 2024a; Li et al., 2024a). As a result, existing methods cannot yet transform system-level evaluation into training signals that are both localized—to the agent and response levels—and auditable (Lowe et al., 2017; Foerster et al., 2018; Lundberg & Lee, 2017; Lightman et al., 2023; Setlur et al., 2024).

This paper makes the following contributions:

- We propose a theoretical framework that transforms system-level evaluations of LLM multi-agent systems into signed, credit-conserving message-level signals, bridging the gap between global outcomes and local supervision.
- We design complementary attribution mechanisms—Shapley-based credit allocation with PRM refinement for successful episodes, and first-error localization with repair-aware preferences for failures—ensuring that both successes and failures yield informative supervision.
- We prove that the resulting signals are bounded, cooperative, and credit-conserving, and show they are directly compatible with reinforcement- and preference-based post-training, providing a scalable path toward reliable training of multi-agent LLM systems.

## 2 BACKGROUND

**Credit assignment and cooperation.** From a deep learning perspective, effective training requires that supervision signals propagate to the parameters responsible for the observed behavior; otherwise, optimization may stall or converge to spurious minima (LeCun et al., 2015; Goodfellow et al., 2016). When multiple agents interact to produce a joint outcome, the analogous challenge arises: improvement depends on correctly identifying which contributions were beneficial, which were detrimental, and their relative magnitudes—often described in terms of *marginal contributions*, *counterfactual impact*, or *credit assignment* (Lowe et al., 2017; Foerster et al., 2018; Ng et al., 1999; Rodrigues et al., 2020). Without such attribution, updates risk being misdirected, leading to suboptimal or unstable learning (Lowe et al., 2017; Foerster et al., 2018; Zhang et al., 2019; Wang et al., 2020). This motivates the need for *credit assignment*, the problem of mapping system-level outcomes back to individual contributors (Sunehag et al., 2018; Rashid et al., 2018; Gronauer & Diepold, 2022; Mahajan et al., 2022).

In reinforcement learning, this problem has been studied extensively in multi-agent settings. Value decomposition methods such as VDN (Sunehag et al., 2018), QMIX (Rashid et al., 2018), and DCG (Böhmer et al., 2020) factorize global value functions into agent utilities and pairwise payoffs, enabling coordinated control. Reward redistribution methods densify sparse or delayed returns: RUDDER (Arjona-Medina et al., 2019) reassigns final outcomes to early timesteps, ARES (Holmes & Chi, 2025) leverages transformer attention for offline shaping, and ABC (Chan et al., 2024) extends redistribution to token-level RLHF. Recent efforts such as MAGRPO (Liu et al., 2025) adapt MARL ideas to LLM collaborations. Together, these works demonstrate that dividing system rewards across agents or timesteps is a powerful driver of cooperative learning. However, they typically assume low-dimensional, repetitive state–action spaces with dense numeric rewards, assumptions that break down in LLM-based MAS where actions are high-dimensional text outputs, responses are rarely

repeated, and evaluations are often sparse, process-based, and non-differentiable (Li et al., 2024b; Yang et al., 2024; Park et al., 2023; He et al., 2025a; Wang et al., 2025b).

**Shapley values and cooperative game theory.** A natural candidate for building evaluation-aligned training pipelines is the *Shapley value*, which links game-theoretic attribution with fair and interpretable distribution of system outcomes (Shapley, 1953; Castro et al., 2009; Maleki et al., 2013). In cooperative game theory, the Shapley value is the uniquely defined solution concept that divides payoffs fairly, satisfying symmetry (equal players receive equal credit), dummy (irrelevant players get zero), and efficiency (credits sum to the total outcome). These axioms have made it a standard attribution rule in economics, political science, and increasingly in machine learning (Lundberg & Lee, 2017; Ghorbani & Zou, 2019).

Formally, for a coalition of players $N$ and a utility function $v : 2^N \to \mathbb{R}$, the Shapley value for player $i \in N$ is:

$$\phi_i(v) = \sum_{S \subseteq N \setminus \{i\}} \frac{|S|!(|N| - |S| - 1)!}{|N|!} \big[ v(S \cup \{i\}) - v(S) \big]. \tag{1}$$

This averages $i$'s marginal contribution across all coalitions, producing an allocation that is order-agnostic and axiomatically fair. Importantly, $\phi_i$ may be negative, meaning the system performs better without that player.

In machine learning, Shapley values underpin explainability (e.g., SHAP (Lundberg & Lee, 2017), TokenShapley (Xiao et al., 2025)), data valuation (Ghorbani & Zou, 2019), and multi-agent RL credit assignment (Li et al., 2021b; Wang et al., 2024). Applied to multi-agent systems, each agent can be treated as a "player," with Shapley allocation attributing outcomes by measuring how much better the system performs with that agent present. This discourages competition and instead rewards **cooperative contribution**.

Recent LLM work has extended Shapley values to token attribution (Zhao et al., 2024; Cao et al., 2025) and coordination among autonomous agents (Hua et al., 2025). However, these efforts focus on attribution alone. The open challenge—and the gap we address—is *how to integrate Shapley-based allocations into post-training pipelines*, turning fair attribution into actionable supervision for improving multi-agent LLM systems.

For additional related work on Shapley values across explainability, data valuation, and multi-agent settings, see Appendix A.1.

**Process reward models.** Process Reward Models (PRMs) extend outcome-based supervision by assigning labels or scores to intermediate steps, providing denser feedback than a single end-of-trajectory reward (Lightman et al., 2023; Wang et al., 2023; Zelikman et al., 2022; Ulmer et al., 2023; Menick et al., 2022). Instead of evaluating only the final answer, PRMs assess whether each reasoning step is valid, enabling models to learn from partially correct traces. Building on this idea, *OmegaPRM* introduces a binary search to locate the *first error* and labels all earlier steps as valid and all subsequent ones as invalid (Luo et al., 2024). This primarily addresses *failure cases*, but leaves success traces trivially marked as fully valid and overlooks inefficiency or redundancy. Moreover, directly applying this "all following steps invalid" assumption to multi-agent workflows is problematic: later agents may attempt to *repair* earlier errors, so judging all subsequent messages invalid unfairly penalizes corrective behaviors.

So far, PRMs and OmegaPRM have been developed only for single-agent reasoning traces (Lightman et al., 2023; Wang et al., 2023; Luo et al., 2024). They also oversimplify success episodes: in chain-of-thought reasoning, a correct chain is labeled with all 1s, but in multi-agent systems even successful runs may contain redundant or irrelevant messages. If all steps are labeled valid, inefficiency and free-riding go unpenalized (Menick et al., 2022; Ulmer et al., 2023). These limitations highlight the need for PRM adaptations that attribute credit more precisely, across both the agent and message levels in multi-agent LLM workflows.

## 3 PROBLEM SETUP

We consider cooperative multi-LLM systems that solve data analysis tasks using role-specialized agents. Concretely, we use a running example with three agents: a *Planner* that proposes analysis

steps, a *Database* agent that issues SQL queries, and an *Analyst* that interprets results. Together they produce an interleaved trajectory of messages leading to a final system output.

**Agents and trajectories.** Let $\mathcal{A} = \{1, \ldots, n\}$ denote the set of agents, each instantiated as a role-specialized policy $\pi_i$ (e.g., different prompt heads or fine-tuned variants of the same underlying FM). At step $t \in \{1, \ldots, T\}$, agent $i_t \in \mathcal{A}$ emits a message $m_{i_t,t}$ in the context of the history prefix $H_{t-1}$. A trajectory is

$$\tau = (H_0, m_{i_1,1}, H_1, m_{i_2,2}, \ldots, H_{T-1}, m_{i_T,T}, H_T),$$

where $H_T$ contains the final system output $y$ (the last message in $\tau$). Messages may be natural language, code, or tool calls; we use "message" and "response" interchangeably.

**System-level evaluation.** Given an input $x$, the system outputs $y$ and an evaluator $\mathcal{E}$ returns a bounded score

$$R_{\text{sys}} \in [0, 1],$$

with $R_{\text{sys}} = 0$ indicating failure and $R_{\text{sys}} > 0$ indicating success (e.g., rubric grade, accuracy, or task reward). Bounding to $[0, 1]$ induces a finite credit pool and discourages runaway incentives. In the Planner–Database–Analyst example, the task may require estimating a scalar statistic or 1D distribution; the evaluator compares the reported value(s) to ground truth, e.g., awarding $R_{\text{sys}} = 1$ for an exact match or $R_{\text{sys}} = 0.85$ for a partially correct estimate.

**Episode formalization.** Each episode is represented as a triple $(x, \tau, R_{\text{sys}})$. For counterfactual analyses, we write $y_S$ for the final output when only agents in $S \subseteq \mathcal{A}$ are active and all others follow a fixed baseline policy $\pi_{\text{base}}$ (e.g., no-op or a frozen reference model). We denote $\mathcal{E}_S$ as the evaluator applied to $(x, y_S)$ and use the canonical mapping $\text{score}(\texttt{fail}) = 0$, $\text{score}(\texttt{success}(r)) = r \in [0, 1]$.

## 4 PROPOSED FRAMEWORK

**Training objectives.** Our framework transforms a single system-level score into localized supervision while preserving cooperation. It aims to (i) **maximize system reward** by attributing the outcome fairly across agents, (ii) **maximize each agent's contribution** by reinforcing marginal (Shapley) impact without degrading others, and (iii) **maximize efficiency** by rewarding informative messages and penalizing redundancy. We realize these goals via two complementary routes: a success route (system → agent → message) and a failure route (first-error localization → preferences).

### 4.1 SYSTEM → AGENT → MESSAGE: SUCCESS-CASE ATTRIBUTION

**Objective.** The success route transforms a *global evaluation signal* $R_{\text{sys}}$ from a successful run into fine-grained, trainable supervision that meets three goals at once. First, it *maximizes system reward* by distributing credit fairly across agents. Second, it *maximizes each agent's contribution* by using Shapley values to measure marginal impact, ensuring that agents are rewarded for what the system achieves because of their presence. Third, it *maximizes efficiency* by refining these agent-level credits into message-level signals with PRM-style supervision, rewarding informative actions while discouraging redundancy. In this way, global outcomes are decomposed into cooperative, actionable feedback that drives both system-level success and efficient local behaviors.

#### 4.1.1 SYSTEM → AGENT: CREDIT DISTRIBUTION VIA SHAPLEY VALUES

System → agent credit distribution refers to the step where the global evaluation signal $R_{\text{sys}}$ (success/failure score or rubric-based outcome) is decomposed into per-agent rewards. In our framework, this route is applied only in the *success case*. We adopt Shapley values as the backbone of this assignment, since they provide a principled and cooperative mechanism for attributing system rewards. The allocation is considered *fair* because it satisfies symmetry (equal agents receive equal credit), dummy (agents with no impact receive zero), and efficiency (credits sum to the total outcome). The Shapley value of an agent reflects its *marginal contribution* to the system's performance, averaged over all possible coalitions of agents. This ensures that credit is tied not to individual performance in isolation, but to how much better the team performs because of an agent's presence.

Formally, for any coalition $S \subseteq \mathcal{A}$, we define its value

$$v(S) \triangleq \text{score}\big(\mathcal{E}_S(x, y_S)\big), \tag{2}$$

where $y_S$ denotes the final system output when only the agents in $S$ are active and all others are replaced by a fixed *baseline policy* $\pi_{\text{base}}$, and $\mathcal{E}_S$ is the evaluator applied to $(x, y_S)$. We use the canonical mapping $\text{score}(\texttt{fail}) = 0$ and $\text{score}(\texttt{success}(r)) = r \in [0, 1]$.

**Simulating coalitions.** For efficiency and stability, we simulate counterfactual coalitions by re-playing the original trace until the first turn of a removed agent; thereafter, agents in $S$ regenerate their messages (with frozen seeds), while absent agents emit baseline outputs (no-op, frozen $\pi_{\text{ref}}$, or masked propagation). This preserves trajectory coherence while ensuring $y_S$ and its score reflect the missing capability.

Given this coalition value function, the Shapley value for agent $i$ is

$$\phi_i = \sum_{S \subseteq \mathcal{A} \setminus \{i\}} \frac{|S|! \, (n - |S| - 1)!}{n!} \Big( v(S \cup \{i\}) - v(S) \Big), \tag{3}$$

which captures the expected marginal contribution of $i$ across all coalitions. We define $\phi_i$ as the *agent reward*—the share of system performance directly attributable to agent $i$. Importantly, $\phi_i$ may be *negative* if the agent's participation lowers the system score; this property ensures that unhelpful or destabilizing behavior is explicitly penalized rather than ignored.

For interpretability, we also define the *credit ratio*

$$\alpha_i \triangleq \frac{\phi_i}{\sum_{j=1}^n \phi_j} = \frac{\phi_i}{R_{\text{sys}}}, \tag{4}$$

which normalizes an agent's Shapley value relative to the total system reward. Finally, the reward for agent $i$ is

$$r_i \triangleq \alpha_i \cdot R_{\text{sys}} = \phi_i, \tag{5}$$

so that

$$\sum_{i=1}^n r_i = R_{\text{sys}}.$$

Thus, Shapley allocation ensures that the system reward $R_{\text{sys}}$ is distributed fairly across agents, with no credit inflation or free-riding. Negative $\phi_i$ values indicate agents whose actions diminish system performance, while the credit ratio $\alpha_i$ records the signed proportion of system value attributable to each agent.

**Preventing competition.** Shapley values reward *unique, cooperative contributions* and average marginal gains over all coalition orderings. Duplicating another agent's work yields near-zero marginal credit, and sabotaging others lowers coalition values—and thus the total pool to divide—so it does not increase one's own share (Shapley, 1953; Castro et al., 2009; Maleki et al., 2013). In the Planner–Database–Analyst example, the Planner gains credit by enabling effective queries, not by reproducing or suppressing the Database's outputs.

**Capturing overall contribution.** Because the Shapley value averages marginal gains across all coalition contexts, it provides a robust measure of an agent's overall contribution to maximizing the system reward $R_{\text{sys}}$. In this sense, it is both *cooperative* (agents are rewarded for helping the team) and *comprehensive* (every coalition context is considered in expectation). The possibility of negative $\phi_i$ values ensures that agents who consistently reduce system quality are explicitly penalized, making the signal both corrective and fair.

### 4.1.2 AGENT → MESSAGE: MESSAGE-LEVEL REWARD ATTRIBUTION

While Shapley values fairly allocate the system reward $R_{\text{sys}}$ to each agent (Sec. 4.1.1), they are *coarse*: all messages of agent $i$ in a trace share the same $r_i = \phi_i$. To obtain *actionable, per-message* signals for training—detecting inefficiency, discouraging redundancy, and avoiding credit hoarding—we refine the agent-level credit into message-level rewards using a PRM-style procedure. Here, a *message* means any agent emission (text, code, or tool call); we use "message" and "response" interchangeably. Importantly, PRM does not alter $r_i$ itself, but only *distributes* it across the agent's messages.

**Signed message labels.** Each agent $i$ acts at indices $\mathcal{T}_i \subseteq \{1, \ldots, T\}$, producing messages $m_{i,t}$ in contexts $H_{t-1}$. A domain-tuned judge $\mathcal{J}$ (LLM-as-judge or compact PRM) provides a discrete label

$$s_{i,t} \in \{-1, 0, +1\}, \tag{6}$$

interpreted as:

- $s_{i,t} = +1$: the message is *aligned* with the agent's overall contribution direction (it pushes the agent further along its path, whether that path is net helpful or harmful to the system);
- $s_{i,t} = -1$: the message is *counter-aligned*, pulling the agent away from its own contribution direction;
- $s_{i,t} = 0$: the message is neutral or irrelevant to the agent's trajectory.

Thus, message labels reflect how a step advances or undermines the agent's own marginal contribution, not the system outcome directly. This separation ensures that if an agent is net harmful ($\phi_i < 0$), then aligned messages ($s_{i,t} = +1$) are penalized most, while counter-aligned ones ($s_{i,t} = -1$) are rewarded for diluting harm.

**From labels to weights.** To normalize across multiple messages, we compute the absolute contribution mass

$$S_i = \sum_{u \in \mathcal{T}_i} |s_{i,u}|, \tag{7}$$

and define allocation weights

$$\omega_{i,t} = \begin{cases} \dfrac{|s_{i,t}|}{S_i}, & S_i > 0, \\ \dfrac{1}{|\mathcal{T}_i|}, & S_i = 0 \quad \text{(uniform fallback)}. \end{cases} \tag{8}$$

**Message-level rewards.** Given an agent-level Shapley credit $\phi_i$, we assign each message a signed share

$$r_{i,t} = s_{i,t}\, \omega_{i,t}\, \phi_i. \tag{9}$$

By construction, credit is conserved:

$$\sum_{t \in \mathcal{T}_i} r_{i,t} = \phi_i, \qquad \sum_{i,t} r_{i,t} = R_{\text{sys}}. \tag{10}$$

**Interpretation.** This attribution yields intuitive behaviors:

- **Helpful agent** ($\phi_i > 0$): aligned messages receive positive rewards, redundant/neutral ones receive near-zero, and counter-aligned ones are penalized.
- **Harmful agent** ($\phi_i < 0$): aligned messages are penalized (as they reinforce harm), while counter-aligned messages are rewarded (as they dilute harm).

Thus, *message-level attribution refines agent credit into actionable signals*, rewarding efficiency and correction while discouraging redundancy and harmful behaviors.

**Connection to single-agent PRM.** Classical PRMs for chain-of-thought (CoT) reasoning label individual steps and reward locally valid ones (Lightman et al., 2023; Wang et al., 2023; Zelikman et al., 2022). Our adaptation differs in two key ways: (i) it *scales* supervision by cooperative contribution ($r_i = \phi_i$), ensuring credit is conserved; and (ii) it allows *signed* allocation, where messages can inherit positive or negative shares depending on both their alignment and the agent's overall contribution. For efficiency, one may reduce to the binary case $s_{i,t} \in \{0, 1\}$, but the signed scheme offers finer control for MAS.

**Clipping and normalization.** Although message-level rewards $r_{i,t}$ are already bounded in practice by normalization ((8)–(10)), extreme Shapley values or noisy judge scores may still cause instability during training. As an optional safeguard, clipping $r_{i,t}$ or $\phi_i$ to a fixed interval (e.g., $[-1, 1]$) or per-episode rescaling (so that $\max_t |r_{i,t}|$ lies in a target range) can improve robustness without altering relative proportions.

## 4.2 System → Agent → Message: Failure-Case Attribution

**Objective.** Failure cases require a distinct route because when $R_{\mathrm{sys}} = 0$, Shapley redistribution offers little actionable guidance: credits must sum to zero, which blurs responsibility and risks penalizing repair attempts alongside true errors. Rather than divide a zero reward, we replace Shapley redistribution with *first-error localization*: pinpointing the earliest harmful message that pushes the trajectory off track. Coupled with task-level judges, this yields preference-based supervision that both isolates the cause of failure and highlights subsequent repair attempts. In this way, failure episodes still advance the same three goals as in success: (i) **maximize system reward** by steering trajectories back toward valid outcomes, (ii) **maximize each agent's contribution** by distinguishing the error-maker from repairers, and (iii) **maximize efficiency** by discouraging unhelpful sprawl after an error. This ensures that even failed traces provide informative and corrective training signals.

**First-error localization.** In failure episodes, we view the trajectory as a sequential trace and search for the *first harmful message* $m_{i^\star,t^\star}$ whose inclusion flips the evaluator's judgment from "still on track" to "failed." This localization is performed efficiently by a binary search over prefixes $H_t$, requiring only $\mathcal{O}(\log T)$ checks. The agent $i^\star$ responsible for $m_{i^\star,t^\star}$ is marked as producing the critical error. Importantly, we do not assume monotone traces where a single error invalidates everything that follows. Instead, later messages are judged in context, allowing messages that attempt to *repair* the trajectory to still receive positive credit.

**Judges for the failure route.** We introduce two complementary judges:

- A *prefix judge* $\mathcal{J}_{\mathrm{pref}}$ detects whether an error has occurred by checking if a partial trajectory is still viable:
$$\mathcal{J}_{\mathrm{pref}}(H_t) \in \{\mathrm{OK}, \mathrm{ERR}\}, \qquad t^\star = \min\{t : \mathcal{J}_{\mathrm{pref}}(H_t) = \mathrm{ERR}\}.$$

- A *failure-alignment judge* $\mathcal{J}_{\mathrm{fail}}$ evaluates messages after $t^\star$ by asking: does this message align with the failed trajectory or counteract it? Given the system input, the failed output, and the current trace, the judge labels
$$q_{i,t}^{\mathrm{task}} = \mathcal{J}_{\mathrm{fail}}(H_{t-1}, m_{i,t}) \in \{1, 0\},$$
where $q_{i,t}^{\mathrm{task}} = 1$ if the message helps steer the trajectory back toward task success, and $0$ if it aligns with the failure outcome.

In practice, these judges can be instantiated via (i) *execution- or constraint-based checks* (e.g., SQL validators, unit tests, schema consistency), (ii) *rubric-based LLM-as-judge* prompts specialized to the task, or (iii) compact PRMs fine-tuned on pairs near $t^\star$. This combination provides both precise error localization and nuanced repair assessment.

**Preference construction.** Once the first error $m_{i^\star,t^\star}$ is localized, we construct contrastive training pairs:
$$\left(H_{t^\star-1}, \; y^+, \; y^- = m_{i^\star,t^\star}\right),$$
where $y^-$ is the harmful message and $y^+$ is a preferred alternative (e.g., from a corrected edit, a successful episode in a similar context, or a human/LLM-provided fix). Following the PRM/OmegaPRM practice of turning valid/invalid judgments into supervision, these pairs yield *preferences* rather than scalar rewards, making them naturally compatible with objectives such as DPO or GRPO.

Together with the success route, this ensures that both successful and failed episodes contribute useful training signals: rewards distribute credit when the system succeeds, while preferences provide corrective guidance when it fails.

## 5 Theoretical Analysis

**Integration into post-training.** Now that we have constructed message-level signals in both success and failure routes, the natural question is: *how can these signals be integrated into post-training?* We view the outputs of our pipeline as learning-ready supervision. In success episodes, the signed, credit-conserving message rewards $\{r_{i,t}\}$ (Sec. 4.1) function directly as dense reward functions for each policy $\pi_i$, enabling reinforcement-learning-based optimization (e.g., actor–critic, PPO/GRPO) at the

*per-message* level. In failure episodes, first-error localization yields contrastive pairs $(H_{t^{\star}-1}, y^+, y^-)$ (Sec. 4.2) that plug seamlessly into preference-based objectives (e.g., DPO/GRPO). In both routes, optional clipping or rescaling (Sec. 4.1.2) improves stability without changing relative proportions.

Readers interested in a more detailed comparison with standard PRM, including explicit differences in how our signals are consumed by post-training pipelines, may refer to the Appendix (Sec. A.2).

**Computational complexity.**    Let $n$ be the number of agents, $T$ the number of turns in a trace, $C_{\text{dec}}$ the average cost to (re)decode a message, and $C_{\text{eval}}$ the cost of a system-level evaluation call.

*Success route (Shapley + PRM):*

- *Exact Shapley:* requires $2^n$ coalition values $v(S)$, each at most one counterfactual run, for $\mathcal{O}(2^n (T\,C_{\text{dec}} + C_{\text{eval}}))$ time.
- *Permutation sampling (practical):* with $M$ sampled permutations, each marginal contribution is estimated once per permutation. Using our *ablation-on-trace* simulator, this costs
$$\mathcal{O}\big(M\,n\,(\bar{T}\,C_{\text{dec}} + C_{\text{eval}})\big),$$
where $\bar{T} \leq T$ reflects early cutoffs; caching prefixes reduces redundancy. Space is $\mathcal{O}(T)$.
- *PRM labeling:* a pass over messages to obtain $s_{i,t}$ and $\omega_{i,t}$ is $\mathcal{O}(T)$ time and $o(T)$ space.

*Failure route (localization + preferences):*

- *First-error localization:* binary search over prefixes is $\mathcal{O}(\log T \cdot C_{\text{eval}})$.
- *Repair scoring/pairs:* scanning suffix $t > t^{\star}$ is $\mathcal{O}(T - t^{\star})$; building $k$ pairs is $\mathcal{O}(k)$.

Overall, our framework is polynomial in $n, T$ under sampling, with tunable $M$ for the credit–compute trade-off. Exponential cost arises only with exact Shapley.

**Theoretical guarantees.**    Our framework satisfies several guarantees:

- *Efficiency / credit conservation:* $\sum_i \phi_i = R_{\text{sys}}$ and $\sum_{i,t} r_{i,t} = R_{\text{sys}}$, so all supervision is budgeted by actual outcomes.
- *Boundedness:* With $R_{\text{sys}} \in [0,1]$ and $\sum_{t \in \mathcal{T}_i} \omega_{i,t} = 1$, each $|r_{i,t}| \leq 1$. Optional clipping/rescaling further stabilizes.
- *Anti-competition:* duplication yields near-zero marginal credit, while sabotage reduces the pool, so neither increases one's share.
- *Repair-awareness:* unlike monotone-invalidating schemes, failure supervision distinguishes errors from repairs, rewarding agents that counteract failures.

**Positioning relative to existing methods.**    Compared with existing approaches: *Standard PRM* provides local step labels but lacks credit conservation and multi-agent grounding—its supervision does not reflect marginal system value. *Raw system-level RL* suffers from sparse and unstable credit assignment across agents. *Monotone first-error PRM* (Omega-style) localizes errors but discards repair signals. Our framework integrates cooperative, Shapley-grounded credits in success and repair-aware preferences in failure, yielding signed, bounded, and conservative signals that are computationally efficient under sampling and directly compatible with modern RLHF/DPO pipelines.

## 6    DISCUSSION AND LIMITATIONS

**Preventing credit hoarding.**    A key risk in cooperative multi-agent systems is that one agent might dominate the workflow, suppressing others to inflate its own marginal credit. Our framework mitigates this through two mechanisms. First, Shapley values enforce efficiency and symmetry, ensuring that duplicated or suppressive behavior yields little marginal gain. Second, message-level PRM supervision distributes an agent's credit across its responses, so redundant or uninformative steps receive near-zero reward even if the agent contributes to the final outcome (Sec. 4.1.2). This combination discourages credit hoarding and aligns incentives with cooperative efficiency.

**Evaluator cost and scalability.** Exact Shapley computation is exponential in the number of agents ($2^n$ coalitions). To scale, we suggest adopting *Monte Carlo permutation sampling*, which estimates

Shapley values by averaging marginal contributions across $M$ sampled agent orderings, yielding $\mathcal{O}(Mn)$ cost per episode. Replay until the first turn of a removed agent further reduces trace length, while prefix caching avoids redundant recomputation. We emphasize that Shapley attribution is best suited for post-training calibration rather than early-stage training. In practice, lightweight surrogates—such as compact models that predict normalized credit ratios $\hat{\alpha}$ or distilled PRM-style judges—can replace full Shapley once bootstrapped. This makes our pipeline feasible even for larger agent teams (Appendix A.3).

**Reliability of judges.** We are also aware of concerns regarding whether human or LLM judges can reliably supervise intermediate steps (Stechly et al., 2024). Our framework reduces this burden: judges need not assess global correctness but only determine if a message is aligned with the agent's intended role (Eq. 6). This lighter form of evaluation lowers the expertise required from humans and reduces systematic LLM errors, since decisions are local and role-specific. Occasional mislabeling only redistributes an agent's own credit across its steps, preserving overall conservation and limiting harm.

**Role-specific assumptions.** Our current framework applies both to homogeneous teams (the same FM role-prompted differently) and heterogeneous teams (specialist FMs such as a domain-specific chemistry model paired with a code translator). In heterogeneous settings, however, careful choice of baseline policies is critical to ensure fair Shapley credit. We leave a fuller treatment of heterogeneous teams and baseline design to Appendix A.5.

**Practical remedies and future work.** Our framework is designed to operate in live settings where episodes and evaluations accumulate during deployment. From these traces, surrogate predictors (compact learned models) can be trained to approximate Shapley credits or response-level judges, significantly reducing cost (Appendix A.4). Attribution fidelity can drift under distribution shift, but periodic recalibration with a small batch of sampled coalitions keeps the surrogates anchored. While we propose a complete theoretical framework, thorough experiments quantifying these trade-offs and efficiency gains are left for follow-up work. Although this paper focuses on theory, our next step is to conduct experiments on LLM MAS benchmarks to empirically validate the framework and compare against existing baselines.

## 7 CONCLUSION

We presented a theoretical framework for LLM multi-agent systems that unifies cooperative game–theoretic attribution with PRM-style supervision, bridging the gap between *global system-level evaluation signals* and *local, trainable supervision*. Our approach transforms global outcomes into *signed, credit-conserving message-level signals* along the full *system → agent → message* pathway: Shapley-based attribution provides a fair division of credit in successful episodes, while first-error localization yields repair-aware preferences in failures. These signals are directly compatible with modern post-training pipelines, enabling reinforcement-learning or preference-based optimization at the per-message level. While efficient approximations (e.g., permutation sampling, surrogate judges) make the framework scalable in practice, future work will extend to heterogeneous specialist teams and empirical validation. We believe this work establishes the conceptual foundation for cooperative, credit-conserving post-training of LLM multi-agent systems.

## ACKNOWLEDGMENTS

The authors used large language models (LLMs) as writing assistants for grammar refinement, formatting, and minor rephrasing. All core ideas, technical contributions, and conceptual development presented in this paper are original to the authors.

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

# A APPENDIX

## A.1 RELATED SHAPLEY LITERATURE

For completeness, we summarize additional strands of work where Shapley values have been applied across economics, political science, and machine learning.

**Foundations in cooperative game theory.** The Shapley value was originally introduced as the unique fair division rule in cooperative games (Shapley, 1953). Variants and computational aspects have since been studied extensively, including polynomial-time approximation methods (Castro et al., 2009; Maleki et al., 2013) and connections to other solution concepts such as the nucleolus (Li et al., 2025b).

**Explainability and attribution in ML.** In machine learning, Shapley values are central to explainability and feature attribution. SHAP (Lundberg & Lee, 2017) popularized their use in practice, while later work extended to token-level (Xiao et al., 2025), document-level, and cluster-level attributions (Ghorbani & Zou, 2019; Ye & Yoganarasimhan, 2025). These methods view each feature, token, or data point as a "player" whose marginal contribution to the prediction or training objective can be quantified.

**Applications in multi-agent and RL.** In reinforcement learning and multi-agent systems, Shapley-style counterfactual credits have been used to allocate returns among agents and measure their influence on a centralized critic (Li et al., 2021b; Zhao et al., 2025; Wang et al., 2025a; 2024). Related directions investigate hierarchical or multi-level settings (Zhao et al., 2025) and speaking/listening teamwork dynamics in ad-hoc agent coordination (Lin et al., 2025b).

**Extensions and generalizations.** Recent work explores generalizations of Shapley-style allocations, such as extending attribution to non-additive or structured outcomes (Bahri et al., 2024; Yang et al., 2025), and adapting cooperative game principles to emerging AI applications. These highlight the broad relevance of Shapley values as a unifying tool for attribution, though they do not directly address integration with post-training pipelines, which is the focus of our framework.

## A.2 INTEGRATION WITH POST-TRAINING

**What standard PRM provides.** Classical Process Reward Models (PRMs) assign step-level binary or continuous labels to intermediate reasoning steps, typically consumed via *weighted supervised fine-tuning* (SFT). Valid steps are up-weighted (e.g., +1), invalid steps are down-weighted or ignored (e.g., 0), yielding a reweighted likelihood objective. This provides useful *local validity* supervision, but it does not guarantee: (i) conservation of credit, (ii) grounding in multi-agent cooperation, or (iii) compatibility with reinforcement learning.

**How our signals differ.** Our framework outputs *signed, credit-conserving* rewards and *repair-aware* preferences:

1. **Signed, credit-conserving rewards.** In success episodes, each message receives a signed reward $r_{i,t}$ with

$$\sum_{t \in \mathcal{T}_i} r_{i,t} = \phi_i, \qquad \sum_{i,t} r_{i,t} = R_{\text{sys}} \in [0,1],$$

   so supervision is *budgeted* by the realized outcome and already shaped like a reward function—making RL-style optimization natural.

2. **Multi-agent grounding via Shapley.** All message signals are scaled by the agent's Shapley credit $\phi_i$, aligning step-level learning with each agent's *marginal contribution* to system performance.

3. **Failure-aware preferences.** When $R_{\text{sys}} = 0$, we localize the first harmful message and construct contrastive pairs $\left( H_{t^\star - 1}, y^+, y^- \right)$ while *still rewarding* subsequent repair attempts—unlike monotone-invalidating schemes that mark all post-error steps invalid.

### A.2.1 PLUGGING THE SIGNALS INTO POST-TRAINING

**Success route (RL-style).** Use $\{r_{i,t}\}$ as per-message rewards for each agent policy $\pi_i$. Standard RLHF-style optimizers (actor–critic, PPO, GRPO) can then be applied at message granularity. For stability: (i) clip $\phi_i$ or $r_{i,t}$ to $[-1, 1]$, or (ii) per-episode rescale so $\max_t |r_{i,t}|$ lies in a target range.

**Failure route (preferences).** From first-error localization, build preference pairs

$$\left( H_{t^\star - 1}, \, y^+, \, y^- = m_{i^\star, t^\star} \right),$$

and train with preference-based objectives (DPO/GRPO). These apply to the error-making agent as well as repair-capable agents whose $y^+$ alternatives demonstrate recovery.

**Hybrid training loop.** A practical loop alternates RL updates on successful episodes with preference updates on failed ones. This unifies both regimes while remaining compatible with existing RLHF/PRM practices.

## A.3 MONTE CARLO APPROXIMATION FOR SHAPLEY VALUES

Exact Shapley computation requires evaluating $2^n$ coalitions, which is infeasible for large $n$. A practical alternative is *Monte Carlo permutation sampling*: draw $M$ random permutations $\pi$ of the agents and compute the marginal contribution of each agent $i$ given the coalition of its predecessors. Formally, for permutation $\pi$ and agent $i$:

$$\text{prec}_\pi(i) = \{ j \in \mathcal{A} : \pi(j) < \pi(i) \}.$$

The estimator is

$$\hat{\phi}_i = \frac{1}{M} \sum_\pi \left[ v(\text{prec}_\pi(i) \cup \{i\}) - v(\text{prec}_\pi(i)) \right].$$

This yields an unbiased estimate of the true Shapley value, with variance decreasing as $M$ grows. In practice, a few hundred samples suffice to stably rank agent contributions, making this approach computationally tractable.

## A.4 SURROGATE CREDIT MODELS AND JUDGES

To further reduce runtime cost, surrogates can be distilled from interaction data:

- *Credit predictors.* Train a compact network $G_\theta(\tau)$ to predict normalized contribution ratios $\hat{\alpha}$ from traces, bootstrapped using sampled Shapley attributions (Castro et al., 2009; Lundberg & Lee, 2017).
- *Process judges.* Train a classifier $J_\phi(H_{t-1}, m_{i,t})$ from human-verified or high-quality LLM labels, following PRM distillation practices (Lightman et al., 2023; Luo et al., 2024; Setlur et al., 2024).

These surrogates amortize attribution and judgment over many runs, providing efficiency while preserving the cooperative grounding of our framework.

## A.5 HETEROGENEOUS TEAMS AND BASELINE POLICIES

Our framework applies both to *homogeneous* teams, where all agents are instances of the same FM role-prompted differently, and to *heterogeneous* teams, where agents are instantiated from different foundation models with specialized capabilities (e.g., a chemistry model, a code-translation model, and a general reasoning model).

In heterogeneous settings, the design of the *baseline policy* $\pi_{\text{base}}$ is especially critical for fair Shapley attribution. Recall that Shapley credit is defined relative to a counterfactual coalition where absent agents are replaced by $\pi_{\text{base}}$. If this baseline is too weak (e.g., random outputs), specialists appear disproportionately valuable; if it is too strong or domain-mismatched, it may suppress legitimate contributions. Thus, baseline design effectively anchors what "absence" means for each role.

Several strategies are possible:

- **Role-conditioned null agents.** Define a minimal but syntactically valid policy per role (e.g., a database agent that always returns an empty table, or a planner that outputs a no-op step). This ensures consistency while avoiding artificial inflation.
- **Skill-matched baselines.** For specialists, use a weaker model of the same type (e.g., a smaller chemistry FM as the baseline for a large chemistry FM), so that credit reflects value *beyond* the basic skill set.
- **Hybrid baselines.** Combine simple heuristics with role conditioning, such as default SQL queries for databases or template-based summaries for analysts, to maintain comparability across domains.

While our current work assumes homogeneous or role-specialized agents with straightforward baselines, extending the framework to fully heterogeneous FM teams requires principled baseline design to prevent distortions in credit assignment. This remains an important avenue for future study, especially as LLM–specialist collaborations become more common.

