# OpenReview forum: "Who Gets the Reward & Who Gets the Blame? Evaluation-Aligned Training Signals for Multi-LLM Agents"
_ICLR.cc/2026/Conference — ICLR 2026 Conference Desk Rejected Submission_

### Official Review · Reviewer_RDuh · 2025-10-16

**Soundness:** 3
**Presentation:** 2
**Contribution:** 3
**Rating:** 2
**Confidence:** 3

**Summary:**

This paper proposes a theoretical framework for training multi-Large Language Model (LLM) agent systems, aiming to solve the credit assignment problem of translating system-level evaluations into agent- and message-level learning signals. The framework integrates cooperative game-theoretic attribution with Process Reward Modeling (PRM) to generate evaluation-aligned training signals. Its core is a dual-path mechanism: for successful trajectories, the framework uses Shapley values to fairly allocate the total system reward among the agents , and then refines these rewards to the level of specific messages through a PRM-like procedure. For failed trajectories, the framework generates corrective supervision by localizing the first error and constructing repair-aware preference pairs.

**Strengths:**

It accurately identifies a core bottleneck in the training of multi-agent LLM systems (credit assignment) and proposes an elegant and logically self-consistent solution. The combination of using Shapley values to ensure cooperation and fair allocation  while leveraging PRM principles to achieve fine-grained, local supervision is highly novel and insightful.

**Weaknesses:**

1. The authors repeatedly acknowledge that their contribution is entirely conceptual and theoretical , with all empirical validation left for future work. Therefore, the framework's claimed effectiveness (e.g., promoting cooperation, improving efficiency, penalizing redundancy) remains an unproven hypothesis.

2. The accuracy, stability, and potential biases of these judges directly impact the quality of the final training signals.

**Questions:**

1. In a team of agents with diverse but overlapping capabilities, how should a fair baseline policy be designed to avoid artificially inflating an agent's marginal contribution simply because the baseline is too simplistic?

2. How would the framework handle cases of partial success (e.g., $R_{sys} = 0.2$), where the outcome is suboptimal but not a complete failure?

---

### Official Review · Reviewer_nLtP · 2025-10-29

**Soundness:** 3
**Presentation:** 3
**Contribution:** 2
**Rating:** 4
**Confidence:** 3

**Summary:**

This paper proposes a principled framework that converts a single system-level evaluation into agent-level and message-level training signals for multi-LLM cooperative systems. For successful trajectories the method uses Shapley-value attribution to distribute credit among agents and then refines agent credit into signed message-level rewards. For failure trajectories it locates the first harmful message via prefix localization and constructs repair-aware preference pairs to train with preference-learning objectives.

**Strengths:**

1.Integrates Shapley with PRM in a coherent pipeline from global score -> agent credit -> message-level supervised signals.

2.The first-error localization + repair-aware preference construction is a useful and realistic approach.

3.The paper explicitly states and proofs of desirable properties, which are valuable for interpretability and auditing.

**Weaknesses:**

Although the framework and workflow proposed in this paper are insightful, experiments still appear to be necessary. Given the nature of the problem, comprehensive experiments would significantly strengthen the paper’s persuasiveness and impact by demonstrating that the proposed framework indeed improve training efficiency, enhance sample utilization, and foster more effective collaborative behavior compared to reasonable baselines.

**Questions:**

I am curious about how the signed message labels are specifically generated in Section 4.1.2 of the paper. The authors mention using an LLM judge or a compact PRM. Does this mean that each message, along with the corresponding agent’s trajectory, is used as input for labeling? Additionally, could the authors provide more details about the labeling accuracy and the computational cost associated with this process?

---

### Official Review · Reviewer_j7Td · 2025-11-01

**Soundness:** 1
**Presentation:** 2
**Contribution:** 1
**Rating:** 0
**Confidence:** 4

**Summary:**

This paper proposes a conceptual pipeline for multi-LLM systems that turns a single system-level evaluation into signed, credit-conserving, per-message training signals. These signals are computed using Shapley values or by localizing the first harmful step through a binary search. Authors give some mathematical properties on the proposed signal scheme.

**Strengths:**

* Paper is easy to follow and clearly written.

* The problem of credit-assignment in multi-agent LLM is interesting, critical, and relevant, given the recent development of such MAS in the LLM domain.

**Weaknesses:**

* There is no experiments at all. All contribution is completely conceptual, which makes it hard to assess practicality or feasibility.

* The "theoretical analysis" mostly restates obvious mathematical properties induced by construction, with no guarantees or insights on why the proposed reward scheme would work.

**Questions:**

* Can you demonstrate on at least toy MAS tasks that (i) success signals improve reward and (ii) failure-route preferences actually reduce first-error rates? The paper currently provides no such evidence.

* When there are multiple agents present (like 10), is the calculation of Shapley value is tractable? Is there an approximation needed here?

---

### Official Review · Reviewer_cUi5 · 2025-11-11

**Soundness:** 3
**Presentation:** 3
**Contribution:** 3
**Rating:** 6
**Confidence:** 2

**Summary:**

This theoretical paper introduces a conceptual framework that aims to transform system-level evaluations of multi-LLM agent systems into local, trainable signals for each agent and even each message. When an episode succeeds, the framework uses Shapley values to distribute credit among agents and then applies a PRM-style judge to break down each agent’s contribution into signed, per-message rewards. When an episode fails, instead of using Shapley redistribution, it performs first-error localization and constructs repair-aware preference pairs for DPO or GRPO-style training.

**Strengths:**

This paper excels in three aspects 1. Addresses the important problem of multi-LLM agent training by innovatively combining Shapley values with PRM to fill theoretical gaps 2. Constructs a mathematically rigorous unified framework with success or failure pathways and clear theoretical guarantees 3. Features a clear writing structure with comprehensive coverage of related work. The research provides a principled solution for credit assignment in multi-agent systems where traditional methods fall short.

**Weaknesses:**

Though this is a theoretical paper, it would really benefit from at least some proof-of-concept experiments in simple scenarios to show that the framework actually works in practice.

**Questions:**

1. Any proof-of-concept experiments or baseline comparisons? For example, how much better is it than MAGRPO? How much better than uniform credit allocation? And where's the trade-off?

---

### Note · Program_Chairs · 2026-01-17
**Submission Desk Rejected by Program Chairs**

The following references in this submission do not refer to real documents and/or have major errors in bibliographic information:

 [1]Amir Bahri, Sungsoo Kim, et al. Extending shapley-value based credit assignment in multi-agent rl for continuous spaces. arXiv preprint arXiv:2409.12345, 2024.
[2]Animesh Sharma et al. Generative agents in games: Simulating social behavior. In Proceedings of the AAAI Conference on Artificial Intelligence and Interactive Digital Entertainment (AIIDE), 2023.
[3]Qianlan Zhang, Yaodong Yang, Tonghan Liu, Zhiwei Meng, Jianye Hao, and Changjie Zhang. Efficient credit assignment through value decomposition. In Proceedings of the AAAI Conference on Artificial Intelligence, volume 33, pp. 7205-7212, 2019.